# Hypoxia-Inducible Factor Prolyl Hydroxylase Inhibitors and Iron Metabolism

**DOI:** 10.3390/ijms24033037

**Published:** 2023-02-03

**Authors:** Chie Ogawa, Ken Tsuchiya, Kunimi Maeda

**Affiliations:** 1Maeda Institute of Renal Research, Kawasaki 211-0063, Japan; 2Biomarker Society, INC, Kawasaki 211-0063, Japan; 3Department of Blood Purification, Tokyo Women’s Medical University, Tokyo 162-8666, Japan

**Keywords:** HIF-PHI, iron metabolism, renal anemia

## Abstract

The production of erythropoietin (EPO), the main regulator of erythroid differentiation, is regulated by hypoxia-inducible factor (HIF). HIF2α seems to be the principal regulator of EPO transcription, but HIF1α and 3α also may have additional influences on erythroid maturation. HIF is also involved in the regulation of iron, an essential component in erythropoiesis. Iron is essential for the organism but is also highly toxic, so its absorption and retention are strictly controlled. HIF also induces the synthesis of proteins involved in iron regulation, thereby ensuring the availability of iron necessary for hematopoiesis. Iron is a major component of hemoglobin and is also involved in erythrocyte differentiation and proliferation and in the regulation of HIF. Renal anemia is a condition in which there is a lack of stimulation of EPO synthesis due to decreased HIF expression. HIF prolyl hydroxylase inhibitors (HIF-PHIs) stabilize HIF and thereby allow it to be potent under normoxic conditions. Therefore, unlike erythropoiesis-stimulating agents, HIF-PHI may enhance iron absorption from the intestinal tract and iron supply from reticuloendothelial macrophages and hepatocytes into the plasma, thus facilitating the availability of iron for hematopoiesis. The only HIF-PHI currently on the market worldwide is roxadustat, but in Japan, five products are available. Clinical studies to date in Japan have also shown that HIF-PHIs not only promote hematopoiesis, but also decrease hepcidin, the main regulator of iron metabolism, and increase the total iron-binding capacity (TIBC), which indicates the iron transport capacity. However, concerns about the systemic effects of HIF-PHIs have not been completely dispelled, warranting further careful monitoring.

## 1. Introduction

Renal anemia is caused by multiple factors, including primarily decreased erythropoietin (EPO) production as the main factor, as well as hematopoietic suppression due to uremia, shortened erythrocyte survival, and abnormal iron metabolism [1]. Renal anemia is the most common complication in end-stage renal failure [2,3] and is arguably the most important complication in patients with chronic kidney disease (CKD) because it affects the activities of daily living as well as renal protection, cardiovascular complications, and prognosis [4,5,6]. The development of recombinant human EPO (rHuEPO) almost 30 years ago dramatically improved the prognosis of renal anemia [7], but it was provided as an injectable formulation and required frequent administration. Subsequently, improved versions of rHuEPO were developed, namely, long-acting erythropoiesis-stimulating agents (ESAs). However, problems remained with ESAs as well, such as concerns about cardiovascular damage due to blood EPO concentrations exceeding the physiological range, ESA resistance, and injectables being the only formulations available [8,9].

Hypoxia-inducible factor prolyl hydroxylase inhibitors (HIF-PHIs) were developed for the treatment of renal anemia via a new mechanism of action [10]. HIF-PHIs are available as oral agents whose primary mechanism of action is to promote endogenous EPO production, and the continuous administration of HIF-PHIs improves anemia at physiological EPO blood concentrations [11,12].

HIF-PHIs also improve iron supply for hematopoiesis [10]. Iron is one of the main components of hemoglobin, and since most of it is supplied via recirculation, ensuring favorable iron metabolism is essential in the treatment of anemia [13,14]. Iron deficiency includes absolute iron deficiency due to malabsorption and so-called functional iron deficiency, a condition in which the body’s iron levels are sufficient but the available iron is insufficient. Both forms of iron deficiency are common in CKD patients with chronic inflammatory conditions. This is due to high blood levels of hepcidin due to inflammation [15]. Many cases of ESA resistance are attributed to iron deficiency [16]. In such cases, oral iron supplementation is not an option, and intravenous (IV) iron supplementation has been used instead [17]. However, non-physiological IV iron supplementation further increases hepcidin and worsens impaired iron utilization, and is also associated with oxidative stress and risk of infection [18,19,20].

HIF-PHIs not only promote endogenous EPO production, but also improve iron supply for hematopoiesis by promoting iron absorption from the intestinal tract and iron recirculation, and, thus, have promise for improving ESA-resistant anemia caused by iron deficiency and reducing the use of IV iron therapy [10].

This article summarizes the current knowledge of iron metabolism, hematopoiesis, and HIF, as well as the effects of HIF-PHIs on iron metabolism and hematopoiesis and the results of clinical studies, with the aim of promoting the effective use of HIF-PHIs.

## 2. Hematopoiesis and Iron Metabolism

Differentiation of the erythroid lineage occurs in the bone marrow, where hematopoietic stem cells differentiate into erythroid progenitor cells such as burst-forming unit-erythroid (BFU-E) and colony-forming unit-erythroid (CFU-E), proerythroblasts, and erythroblasts. These cells then undergo denucleation to become reticulocytes. After being released into the peripheral blood, the reticulocytes differentiate into mature erythrocytes within 1 to 2 days [21]. Erythropoietin receptor (EPOR) is expressed on erythroid progenitor cells and proerythroblasts, and binding of EPO to it inhibits the apoptosis of CFU-E and erythroblasts and promotes their differentiation and proliferation [22,23]. In this process, erythroferon (ERFE) is produced by erythroblasts, which suppresses hepcidin production and promotes hemoglobin synthesis [24] (Figure 1). Therefore, erythropoietin and iron, which is the main component of Hb, are essential for hematopoiesis. Moreover, erythropoietin production is mainly regulated by HIF2α [25].

An estimated 25 mg of iron is used for hematopoiesis per day, which is 5–6 times the amount of iron present in the blood. Most of this iron is supplied by recirculation from reticuloendoplasmic macrophages. When erythrocytes become old, they are phagocytosed and processed by reticuloendoplasmic macrophages, and the divalent iron derived from hemoglobin is returned to the blood via the ferroportin (FPN) of reticuloendoplasmic macrophages [13,14]. The divalent iron transported into the blood is oxidized to trivalent iron by ceruloplasmin [26], binds to transferrin (Tf), and is taken up again by transferrin receptor1 (TfR1)-expressing proerythroblasts and erythroblasts (Figure 2). TfR1 expression peaks in the basophilic and polychromatophilic stages [27], and most of the iron taken up in erythroid cells is used for hemoglobin synthesis. Because this process requires a large iron supply, it is speculated that iron-loaded endosomes come into direct contact with mitochondria (kiss and run) [28] and that this process involves transporter mitoferrin 1 found in the mitochondrial inner membrane [29].

Furthermore, iron is thought to be an independent regulator of hematopoiesis during early differentiation [30]. In fact, evidence suggests that apoptosis is enhanced in early hematopoiesis during iron deficiency [31].

## 3. Merits and Demerits of Iron

Organisms have acquired an efficient mechanism of aerobic energy production, and central to this mechanism is oxygen transport, which is carried out by iron in hemoglobin [32]. Iron is also involved in energy production, detoxification of reactive oxygen species, and DNA synthesis, and is an essential ion that plays a fundamental role in the maintenance of life. Iron deficiency results in impaired cardiac function, suppressed immune cell production, cerebral nerve damage, and immunodeficiency [33]. At the same time, iron ions are highly toxic and, thus, are strictly controlled within living organisms. The physiologically active form of iron is divalent iron, and its excess presence in cells induces hydroxyl radical production via the Fenton reaction [34], leading to carcinogenesis [35] and cardiovascular [36,37] and endocrine disorders. Excess iron also leads to increased susceptibility to infection [38] and intracellular sequestration of iron [39]. This is because most pathogens use iron for survival and growth, and iron signaling increases hepcidin, the main regulator of iron metabolism. Therefore, it is important to control iron levels without excess or deficiency while taking into account the state of iron metabolism.

Recently, IV iron supplementation has frequently been used to treat iron deficiency, especially functional iron deficiency in hemodialysis patients. Although IV iron supplementation provides a transient supply of iron [40], most of the iron supplied enters the iron stores and does not lead to the retention of available iron, forcing patients to repeat IV iron supplementation. Recently, there have been reports that high-dose IV iron therapy is beneficial for cardiac function and improves the prognosis of heart failure and hemodialysis patients [41,42]. However, these are all short-term studies, and concerns remain regarding the long-term prognosis. Low transferrin saturation (TSAT), which indicates decreased available iron, regardless of serum ferritin and hemoglobin levels, has also been linked to cardiovascular events and poor prognosis, suggesting the need for good iron metabolism.

## 4. Iron Metabolism

### 4.1. General Information

The total amount of iron in the human body is 3 to 4 g. A very small amount of iron (1–2 g/day) is absorbed via the intestinal tract, while the same amount is lost from the body due to the shedding of the intestinal epithelium. Thus, there is no active iron excretion mechanism, and the body’s iron balance is maintained by regulating iron absorption. Due to this “semi-closed circuit”, most of the body’s iron requirements are met by recirculation. More specifically, when blood iron levels are insufficient, iron is supplied from iron-storing ferritin. In the body, about 70% of iron is distributed in the reticuloendothelial system and about 30% in the liver as ferritin iron stores. These two systems account for almost all of the iron in the body, with only 2–4 mg of iron in the blood [43]. Iron supply to the blood is mainly provided by intestinal cells, reticuloendothelial macrophages, and hepatocytes via FPN; thus, it is the amount of FPN that determines iron supply [44]. Furthermore, when hepcidin binds to FPN, the conjugate is degraded in lysosomes, making hepcidin the main regulator of iron metabolism [14] (Figure 2).

Hepcidin is a peptide produced in the liver. Although the system of hepcidin produc- tion has not yet been fully elucidated, the molecule is believed to be regulated mainly by the BMP/SMAD (bone morphogenetic protein/small-body-size mothers against decapentaplegic homolog 1) system, which senses and regulates iron, and the IL-6/STAT3 (interleukin-6/signal transducers and activators of transcription 3) system, which is initiated by inflammatory signals [45,46]. Recent evidence suggests that ERFE suppresses hepcidin production by directly acting on the BMP/SMAD system [47].

### 4.2. Iron Absorption from the Intestinal Tract

Iron absorption into the body occurs mainly in the duodenum and upper jejunum via the non-heme and heme iron pathways [48]. Plant-derived non-heme iron is reduced from trivalent to divalent by duodenal cytochrome B (DCYTB) in the duodenal epithelium [49] and is taken up by intestinal cells via divalent metal transporter 1 (DMT1) [50]. On the other hand, the absorption pathway of animal-derived heme iron has not yet been elucidated. Heme iron has been reported to be taken up via heme carrier protein 1 (HCP 1) [51], but other reports suggested the involvement of a folate transporter [52], indicating some controversy. Heme iron taken up into cells is reduced to divalent iron by hemoxygenase [53]. When the body needs iron, it is supplied to the blood via FPN, an iron-transporting membrane protein located on the vascular side of intestinal cells [54]. Then, iron is oxidized to trivalent iron by membrane-bound hephaestin or its homolog soluble ceruloplasmin [26,55] and binds to Tf, which ensures safe transport [56]. When iron is not needed, on the other hand, it remains stored in ferritin in intestinal cells and is excreted from the body by the shedding of the intestinal epithelium (Figure 3a).

### 4.3. Iron Uptake into Cells

Iron binds to TfR1 expressed on the cell membrane surface and is taken up into cells by endocytosis [56]. Iron is then incorporated into endosomes, where it is reduced to divalent iron by STEAP3 (6-transmembrane epithelial antigen of the prostate 3) [30], and then transported into the cytoplasm via DMT1, an endosomal membrane protein [57]. When iron is needed immediately, it is transported to mitochondria and used for hemoglobin synthesis, energy production, DNA synthesis, and other processes, whereas when iron is not needed immediately, it is stored in ferritin and used when needed. Iron uptake into cells is mainly through the Tf-based pathway, and it is estimated that almost all iron is taken up by this pathway in immature erythroid cells [56]. However, iron transport by other pathways, such as non-Tf-bound iron transporters such as ZRT/IRT-like protein 14, DMT1, and ferritin, has also been identified in some cells, including hepatocytes [58,59,60] (Figure 3b).

### 4.4. Intracellular Iron Regulation

The intracellular labile iron pool (LIP) generates hydroxyl radicals via the Fenton reaction [34] and, thus, is strictly regulated in cells by regulating the expression of proteins involved in intracellular iron transport. This regulation mechanism is known as the iron responsive element (IRE)–iron regulatory protein (IRP) system, in which gene translation is regulated by the binding of IRPs to IREs located in mRNAs. When an IRP binds to an IRE in the 5′ untranslated region (5′ UTR), translation is inhibited [61], whereas when it binds to an IRE in the 3′ UTR, mRNA stability is increased and translation is enhanced [62]. IRPs are activated under iron deficiency only, meaning that the production of proteins that have IREs in the 5′ UTR is suppressed, such as ferritin, FPN, ARAS2 (erythroid-specific delta-aminolevulinate synthase 2), which is an enzyme involved in the initial steps of heme synthesis, and ACO2 (mitochondrial aconitase 2), which is involved in mitochondrial energy production [61,62,63,64,65,66]. By contrast, production is increased for the TfR1 and DMT1, which have IREs in the 3′ UTR [62]. HIF2α has also been reported to have an IRE in the 5′ UTR and to be under the control of IRP [62,67] (Figure 4).

Intracellular ferritin-mediated regulation of the LIP has also recently been elucidated. When bound to PCBP (poly(rC)-binding protein), which has been implicated in RNA stabilization and translation regulation, iron is transported and stored in ferritin, resulting in a decrease in intracellular LIP [68]. On the other hand, upon binding to NCOA4 (nuclear receptor co-activator-4), ferritin is degraded in lysosomes (ferritinophagy), releasing stored iron and increasing intracellular LIP [69] (Figure 3b).

## 5. HIF

### 5.1. General Information

HIF is a transcription factor that stabilizes under hypoxic conditions and is composed of two units, 1α, 2α, or 3α and β. The β unit is always present, while the α unit is subject to regulation by oxygen concentration [70]. When the α unit is hydroxylated by PHD1-3 (prolyl hydroxylase domain 1-3), it binds to VHL (von Hippel Lindau) protein, is ubiquitinated, and is degraded by proteases [71,72]. The fraction of α units hydroxylated by PHD depends on oxygen concentration. Under normoxic concentrations, hydroxylation proceeds, resulting in very few HIFs, whereas under hypoxia, PHD is inactivated and HIF is stabilized. The binding of this HIF to the enhancer of the target gene promotes transcription, resulting in the expression of the target gene. HIF1α is ubiquitous, whereas HIF2α is localized to renal peritubular interstitial cells, endothelial cells, hepatocytes, cardiomyocytes, glial cells, and type-II pneumocytes [73,74,75,76,77,78]. Nevertheless, there are reported to be hundreds or more target genes for HIFs [79,80,81].

PHD is an iron-dependent enzyme, and its activity, therefore, seems to be attenuated at low iron concentrations, resulting in the stabilization of HIF [82].

### 5.2. HIF and Hematopoiesis

Gene expression of the hematopoietic hormone EPO is regulated by HIF [25]. EPO production takes place in the liver during the fetal and early postnatal periods, but thereafter in the kidneys [83,84,85]. EPO production from the liver also occurs under moderate to severe hypoxic conditions but is not enough to improve anemia under renal dysfunction or inflammatory conditions [86,87]. EPO production in the kidneys takes place in fibroblast-like cells in the tubulointerstitium at the corticomedullary border, which are referred to as renal EPO-producing (REP) cells [88,89]. Normally, very few active REP cells (ON-REP cells) are present, accounting for only 10% of all REP cells even under severe chronic anemic conditions, and most REP cells are inactive (OFF-REP cells) [90,91]. This may indicate that these cells have a considerable functional reserve in preparation for more severe conditions. In ON- and OFF-REP cells, EPO gene expression is strictly regulated by HIF in an oxygen concentration-dependent manner [87,90,92]. The kidney is an organ with a high oxygen demand and is prone to hypoxia due to the presence of arteriovenous communications in the tubulointerstitium. Previous studies have shown that blood flow in the renal cortex reflects oxygen consumption [93], suggesting that the kidney is the optimal site for the regulation of EPO production by HIF. Although PHD2 and HIF2α are important for EPO production in the kidneys [94,95], PHD1-3 must be inactivated in the liver [96], suggesting the organ specificity of PHDs involved in EPO production. HIF1α has been shown to induce the expression of GATA1, an important factor in the differentiation of erythroid cells, and to inhibit the apoptosis of CFU-E and erythroblasts [97] (Figure 1). Recent studies have also indicated the involvement of HIF3α in EPO production [98,99].

Furthermore, it has been reported that HIF upregulates EPOR expression in cancer cells and nerve cells [100,101], and this may also be the case in hematopoietic cells.

### 5.3. HIF and Iron Metabolism

HIF is also closely involved in the regulation of proteins and enzymes involved in iron metabolism. First, it suppresses the production of hepcidin, the main regulator of iron metabolism, and promotes FPN synthesis. The hepcidin-inhibitory effect of HIF was initially considered to be a direct action [102,103], but is now considered to be an indirect action via ERFE [104,105] (Figure 1). It has also been reported that FPN synthesis is enhanced by the direct action of HIF2α [106]. These observations suggest that HIF increases iron supply from cells to the blood. Furthermore, with regard to iron absorption from the intestinal tract, studies have shown that HIF2α promotes the synthesis of DMT1 and DCYTB [107,108] and HIF1α promotes the synthesis of hemoxygenase [109], suggesting that HIF is also involved in iron absorption from the intestinal tract (Figure 3a). HIF1α has also been described to induce ceruloplasmin, Tf and TfR1 [110,111,112], implying that HIF also promotes iron transport in the blood and iron uptake into hematopoietic cells (Figure 2).

NCOA4 (nuclear receptor coactivator 4), which is involved as a cargo receptor in ferritinophagy, the process by which iron stored in intracellular ferritin is returned to the LIP, has also been described to be induced by HIF2α, suggesting that iron supply from iron stores is also enhanced [113] (Figure 3b).

## 6. HIF-PHI

### 6.1. General Information

HIF-PHIs were developed to target EPO, one of the target genes for the treatment of renal anemia. In CKD, fibrosis of the interstitium occurs as the disease progresses [114,115], leading to the transformation of REP cells into myofibroblasts (MF-REP cells) and reduced EPO production, resulting in renal anemia [116]. However, MF-REP cells are reversible [89]. Recent evidence has suggested that the suppressed gene expression of EPO in MF-REP cells involves the over-activation of PHD in the early phase and methylation of the gene promoters of EPO and HIF2α in the late phase [91,117]. PHD belongs to a large family of 2-oxoglutarate (2OG)-dependent dioxygenases. The mechanism of action of HIF-PHIs is to suppress PHD2 activity by the structural analogs of 2OG, stabilize HIF2α, and induce quiescent REP cells to express the EPO gene [118]. Therefore, HIF-PHIs may become less effective as renal impairment progresses.

HIF-PHIs also appear to have an effect on iron metabolism via HIF, and, in fact, clinical data have shown a reduction in hepcidin and elevation in TIBC, which reflects Tf levels. Available iron deficiency is a common cause of ESA resistance [16], and HIF-PHIs are expected to be effective in patients with these conditions as well.

### 6.2. Efficacy of HIF-PHI in the Clinical Setting

Since the launch of roxadustat in November 2019, several HIF-PHI products have been launched in Japan, with five products currently available for renal anemia: roxadustat, vadadustat, daprodustat, enarodustat, and molidustat. The effects of HIF-PHIs on hematopoiesis and iron metabolism in the clinical setting are summarized based on the results of the phase 3 trials in Table 1 and Table 2 [119,120,121,122,123,124,125,126,127,128,129,130].

In terms of dosage, only roxadustat was administered three times per week, while the other four products were administered once daily. For all HIF-PHI products, the dose was fixed for the first 4 weeks of treatment, after which the dose was adjusted according to the Hb level. For iron management, only ESA-naïve patients on roxadustat received IV iron supplementation when the condition “TSAT < 5% or serum ferritin (s-ft) < 30 ng/mL” was met, while all other patients were managed to maintain “TSAT ≥20% or s-ft ≥100 ng/mL” according to the guidelines of the Japanese Society for Dialysis Therapy.

Data at weeks 4 and 52 (week 24 for enarodustat only) showed that only the total iron-binding capacity (TIBC), which reflects Tf levels, was elevated from week 4 in all studies. In contrast, hepcidin levels were decreased in all groups at Week 52, although some of the patients with decreased Hb showed an increase in hepcidin levels at Week 4.

Although iron deficiency associated with HIF-PHI treatment has been a concern, the results at Week 4 showed an increase in serum ferritin in patients with decreased Hb. These results suggest that iron demand is strongly influenced by hematopoiesis and is not necessarily increased by HIF-PHI treatment. The hepcidin-inhibitory effect of HIF-PHI is thought to be an indirect effect via ERFN. Thus, a reduced hematopoietic response may also attenuate hepcidin inhibition by HIF-PHIs.

Studies of vadadustat and enarodustat showed an increased mean corpuscular volume (MCV) and mean corpuscular hemoglobin (MCH), suggesting an improved iron uptake into erythrocytes [124,125,127]. Studies of roxadustat and molidustat showed that mildly elevated C-reactive protein (CRP) levels (cutoff, 0.3 mg/dL) had little effect on their dosage and Hb control [120,131,132], while studies of daprodustat showed that its dosage was not influenced by the erythropoietin resistance index (ERI) [122,123]. During inflammation, available iron deficiency due to increased hepcidin levels results in ESA resistance. HIF-PHIs may improve iron supply from macrophages and hepatocellular into the plasma even in ESA-resistant patients through actions such as lowering hepcidin levels. In iron management during HIF-PHI use, attention should be paid to the TSAT value. TSAT is calculated as serum iron divided by TIBC. Therefore, an increase in TIBC due to HIF-PHI use results in a lower TSAT value even if the serum iron level is the same. We examined hematopoiesis- and iron-related parameters 4 weeks after switching from darbepoetin alfa to roxadustat. The results showed enhanced hematopoiesis, suppressed hepcidin, and increased TIBC, immediately after the switch, but on Day 28, the TSAT value was approximately 4.4% lower in the roxadustat group than in the darbepoetin alfa group with comparable serum iron levels [133]. TIBC is elevated not only by HIF1α but also by iron deficiency. However, in our study, despite the low median serum ferritin level of 46.6 mg/dL at baseline, the increase in TIBC was not associated with changes in serum ferritin. It positively correlated with only the change in the red blood cell count [134], suggesting that the increase in TIBC may be due to the effect of HIF. During HIF-PHI use, it may be better to use the serum iron level to determine the amount of available iron.

The only HIF-PHI currently on the market worldwide is roxadustat, which is available in the EU, UK, China, South Korea, and Chile. Global phase 3 data are available for roxadustat [135,136,137,138,139,140,141,142], vadadustat [143,144], and daprodustat [145,146,147,148]. Molidustat has been studied in phase 2 trials [149,150], but has not yet entered phase 3 trials. Enarodustat is under development in South Korea and China, but no clinical data have yet been reported outside of Japan (Table 3).

Serum ferritin levels in the Japanese and global phase 3 clinical trials were approximately 100–150 ng/mL and 250–550 ng/mL, respectively, in non-dialysis-dependent (NDD) patients and approximately 90–140 ng/mL and 440–1000 ng/mL, respectively, in dialysis-dependent (DD) patients. Similarly, hepcidin levels were approx. 40–60 ng/mL and 40–160 ng/mL in NDD patients and 30–70 ng/mL and 130–280 ng/mL in DD patients, respectively. Even taking into account the differences in measurement systems, there were large discrepancies between the Japanese and global studies [119,120,121,122,123,124,125,126,127,128,129,130,135,136,137,138,139,140,141,142,143,144,145,146,147,148,149,150]. Even in this population of patients with high ferritin and hepcidin levels, however, treatment with roxadustat, vadadustat, and daprodustat all showed comparable efficacy in increasing Hb, lowering hepcidin, and increasing TIBC compared to ESAs [140,141,145,146]. For roxadustat, a hematopoietic effect unaffected by mild CRP elevation and reduced IV iron use have also been reported [139].

### 6.3. Safety Considerations

HIF-PHIs continuously increase HIF concentration and, thus, are associated with concerns about systemic effects, such as negative effects on angiogenesis, glucose/lipid/mitochondrial metabolism, cell proliferation, and regulation of matrix turnover and fibrogenesis. Safety studies of HIF-PHIs are, therefore, being conducted with attention to the development of cardiovascular disorders, thrombosis, malignancy, retinopathy, and renal impairment.

A pooled analysis of data from the global phase 3 trials of roxadustat demonstrated safety with respect to cardiovascular events and prognosis [151,152]. A pooled analysis of the Japanese phase 3 trials of daprodustat also showed no increase in thromboembolic, retinal, cardiovascular, and malignancy events compared to the ESA group [153]. Other data have also shown that vascular endothelial growth factor (VEGF) was not increased at week 52 of treatment [130,154] and that the decrease in eGFR did not deviate from the range of natural history [128,129,154].

Safety meta-analyses have also been conducted using data on roxadustat, daprodustat, and other HIF-PHIs. The majority of the results show that the incidence of serious adverse events with HIF-PHIs is similar to that with a placebo or ESAs, although some have shown increased or decreased incidence [155,156,157,158,159,160]. Most recently, a meta-analysis of studies of daprodustat showed a lower incidence of major adverse cardiovascular events (MACE) in patients with DD-CKD in comparison with rHuEPO [161]. However, none of these studies were long-term, and the safety of HIF-PHIs needs to be further evaluated.

## 7. Conclusions

As described in this article, the organism senses oxygen levels acutely in the kidneys and regulates oxygen-carrying erythrocytes. EPO, HIF, and iron are closely interrelated, and their crosstalk maintains oxygen supply to the organism. HIF-PHIs, supplied as oral drugs, exert HIF-mediated hematopoietic and iron metabolism-enhancing effects in the presence of physiological concentrations of EPO, and, thus, have the potential to overcome the negative aspects of ESAs. However, concerns remain regarding the systemic effects of HIF-PHIs, warranting further investigation of their long-term safety.

## Figures and Tables

**Figure 1 ijms-24-03037-f001:**
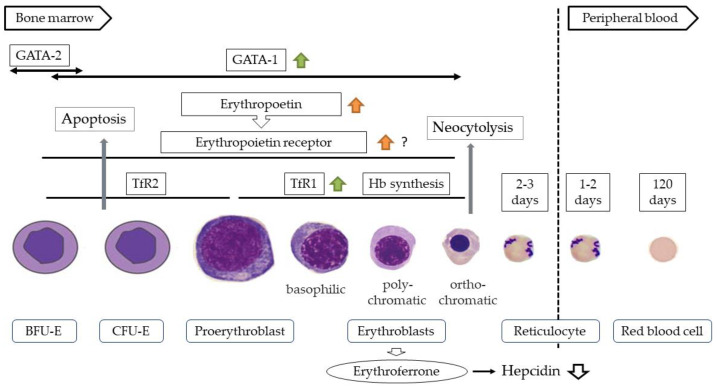
Differentiation process of erythroid cells. BFU-E, burst-forming unit-erythroid; CFU-E, colony-forming unit-erythroid. 
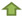
, effect of HIF1α; 
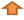
, effect of HIF2α; 
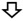
, indirect effect of HIF2α.

**Figure 2 ijms-24-03037-f002:**
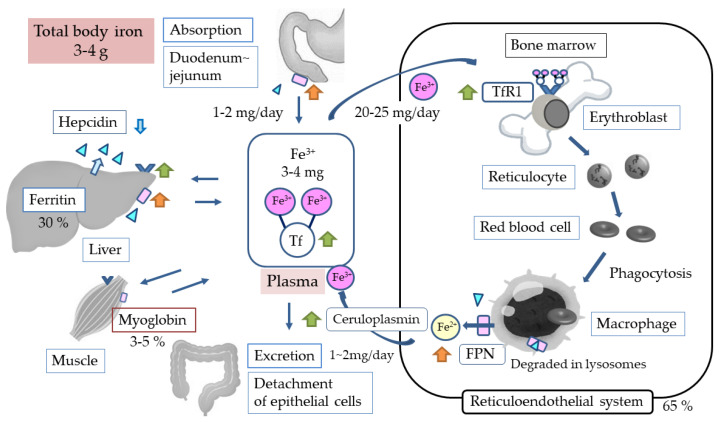
General iron metabolism. Tf, transferrin; TfR1, transferrin receptor1 (
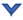
); FPN, ferroportin (
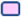
). 
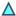
, Hepcidin; 
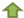
, effect of HIF1α; 
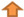
, effect of HIF2α; 
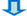
, indirect effect of HIF2α.

**Figure 3 ijms-24-03037-f003:**
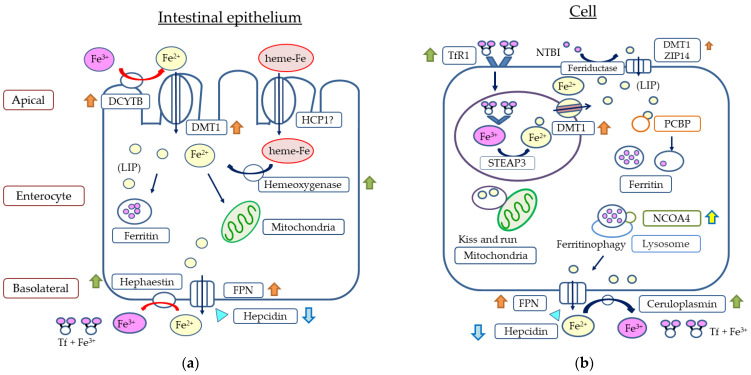
Intestinal epithelium (**a**) and cellular iron metabolism (**b**). DCYTB, duodenal cytochrome B; DMT1, divalent metal transporter 1; HCP1, heme carrier protein1; FPN, ferroportin; LIP, labile iron pool; Tf, transferrin; TfR1, transferrin receptor 1 (
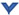
); ZIP14, ZRT/IRT-like protein 14; PCBP, poly(rC)-binding protein; NCOA4, nuclear receptor co-activator; STEAP3, 6-transmembrane epithelial antigen of the prostate; NIBI, non-transferrin-bound iron; 
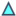
, Hepcidin; 
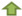
, effect of HIF1α; 
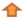
, effect of HIF2α; 
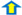
, effect of HIF1α + HIF 2α; 
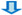
, indirect effect of HIF2α.

**Figure 4 ijms-24-03037-f004:**
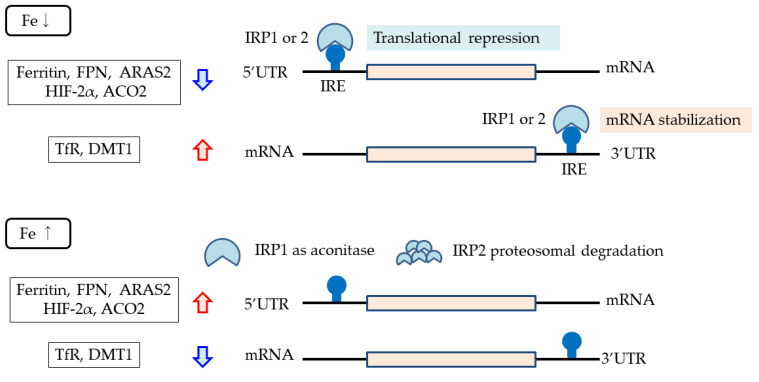
IRE–IRP system. When an IRP binds to an IRE in the 5′ untranslated region (5′ UTR), translation is inhibited, whereas when it binds to an IRE in the 3′ UTR, mRNA stability is increased and translation is enhanced. IRPs are activated and bind to IREs only under iron deficiency. Proteins with IREs in the 5′ UTR include ferritin, ferroportin, ARAS2, and ACO2. Proteins with IREs in the 3′ UTR include transferrin receptor and DMT1. ARAS2, erythroid-specific delta-aminolevulinate synthase; ACO2, mitochondrial aconitase 2; IRE, iron-responsive element; IRP, iron regulatory protein.

**Table 1 ijms-24-03037-t001:** Change from baseline in Hb and iron-related factors during HIF-PHI therapy in non-HD patients.

		Roxadustat		Daprodustat		Vadadustat		Enarodustat		Molidustat	
Starting dose		3 times/wk		Once daily		Once daily		Once daily		Once daily	
		ESA (−); 50 mg or 70 mg *		4 mg		300 mg		2 mg		ESA (−): 25 mg	
		ESA (+); 70mg ** or 100mg ***								ESA (+); 25mg ^†^ or 50 mg ^††^	
Maintenance dose (/day)		20–300 mg (or ≤3 mg/kg)		1–24 mg		150–600 mg		1–8 mg		5–200 mg	
ESA		(−)	(+)	(−)	(+)	(−)	(+)	(−)	(+)	(−)	(+)
Wk 4	Hb	↑	↑	↑	↑	↑	→	↑	↓	↑	↓
	s-ferritin	↓	↓	↓		↓		↓		↓	↓
	s-iron	-	↓	↑		-		-		→	↓
	TIBC	↑	↑	↑		↑		↑		↑	↑
	TSAT	↓	↓	↓		↓		↓		↓	↓
	Hepcidin	↓	↓	↓		↓		↓		-	-
	transferrin	↑	↑	-		-		-		-	-
Wk 52 (24 ^#^)	s-ferritin	↓	↓	↓		↓		↓^#^	↓^#^	↓	↓
	s-iron	↑	↓	→		-		↑^#^	→^#^	↑	↓
	TIBC	↑	↑	↑		↑		↑^#^	↑^#^	↑	↑
	TSAT	→	↓	↓		→		↑^#^	↓^#^	↑	↓
	Hepcidin	↓	↓	↓		↓		↓^#^	↓^#^	↓	↓
	transferrin	↑	↑	↑		-		-	-	-	-
Mean or median ^#^ Dose (mg/day)		50 mgG; 36.3 70 mgG; 36.8	44.2	4 ^#^	5.3 ^#^	335.65	403.67	2.68		46.3	51.21
Evaluation period		wk22	wk 48	wk 40–52		Wk 52		wk 0–24		wk 0–52	
Ref		119	120	122		124		126		128	129

ESA, erythropoiesis-stimulating agent; Hb, hemoglobin; TIBC, total iron-binding capacity, which reflects transferrin levels; TSAT, transferrin saturation. ESA (-), ESA Naïve; ESA (+), User. *; independent of body weight (divided by the web registration system), **; ESA dose before switching [darbepoetin alfa (DA) < 20 μg/wk, continuous erythropoietin receptor activator (CERA) ≤ 100 µg/4 wk, recombinant human erythropoietin (rHuEPO) < 4500 IU/w)], ***; ESA dose before switching (DA ≥ 20 μg/wk, CERA > 100 µg/4 wk, rHuEPO ≥ 4500 IU/wk), ^†^; ESA dose before switching (DA ≤ 30 μg/4 wk, CERA 25 μg/4 wk, rHuEPO ≤ 3000/2 wks), ^††^; ESA dose before switching (DA > 30 μg/4 wk, CERA 25 μg/4 wk, rHuEPO > 3000/2 wks); ^#^, Evaluation at 24 weeks.

**Table 2 ijms-24-03037-t002:** Change from baseline in Hb and iron-related factors when switching from ESA to HIF-PHI in HD patients.

		Roxadustat	Daprodustat	Vadadustat	Enarodustat	Molidustat
Starting dose (/day)		Same as in NDD patients	Same as in NDD patients	Same as in NDD patients	Once daily 4 mg	Once daily 75 mg
Maintenance dose (/day)					Same as in NDD patients	Same as in NDD patients
Wk 4	Hb	↑	↓	↓	↑	↓
	s-ferritin	↓	↑	↑	↓	↑
	s-iron	↓	↑	-	↑	↑
	TIBC	↑	↑	↑	↑	↑
	TSAT	↓	↑	↑	↑	↑
	Hepcidin	↓	↓	↑	↓	-
	transferrin	↑	-	-	-	-
Wk 52 (24 ^#^)	s-ferritin	↓	↓	→	↑^#^	↑
	s-iron	→	↑	-	↓^#^	↑
	TIBC	↑	↑	↑	↑^#^	→
	TSAT	↓	→	→	→^#^	↑
	Hepcidin	↓	↓	↓	↓^#^	↓
	transferrin	↑	↑	-	-	-
Mean or median ^#^ Dose (mg/day)		74.1 (43.6)	ERI < 4.79; 4.0 ^#^ ERI (≥4.79, <8.0); 6.0 ^#^ ERI ≥ 8.0; 6.0 ^#^	367.65	4.26	79.02
Evaluation period		wk 50	wk 40–52	wk 52	wk 20–24	wk 0–50
Ref		121	123	125	127	130

ERI, erythropoiesis-stimulating agent resistance index; Hb, hemoglobin; NDD, non-dialysis dependent; TIBC, total iron-binding capacity, which reflects transferrin levels; TSAT, transferrin saturation; ^#^, Evaluation at 24 weeks.

**Table 3 ijms-24-03037-t003:** Global studies of HIF-PHI.

	Trials	n	ESA	Comparator	Period (wks)	Ref
Roxadustat	phase 3					
NDD	ALPS	594	Naïve	Placebo	52	[135]
	DOLOMITES	616	Naïve	DA	104	[136]
	ANDES	922	Naïve	Placebo	52–234	[137]
	OLYMPUS	2781	Naïve	Placebo	52–208	[138]
Incident DD	HIMALAYAS	1043	Naïve	Epoetin alfa	52	[139]
DD	ROCKIES	2133	Naïve	Epoetin alfa	52	[140]
			User			
	SIERRAS	741	User	Epoetin alfa	52	[141]
	PYRENEES	836	User	DA or epoetin alfa	52–104	[142]
Vadadustat	phase 3					
NDD	PRO_2_TECT	1751	Naïve	DA	52	[143]
		1725	User	DA	52	
Incident DD	INNO_2_VATE	369	User	DA	52	[144]
DD		3554	User	DA	52	
Daprodustat	phase 3					
NDD	ASCEND-ND	3872	Naïve	DA	52	[145]
DD	ASCEND-D	2964	User	Epoetin alfa (HD)	52	[146]
				Subctaneous DA (PD)		
	ASCEND-ID	312	Limited user	DA	52	[147]
	ASCEND-TD	407	User	Epoetin alfa	52	[148]
Molidustat	phase 2b					
NDD	DIALOGUE 1	121	Naïve	Placebo	16	[149]
	DIALOGUE 2	124	User	DA	16	
HD	DIALOGUE 4	199	User	Epoetin alfa/beta	16	
	Extension Studies					[150]
NDD	DIALOGUE 3	83	Naïve	Placebo	36	
		77	User	DA	36	
HD	DIALOGUE 5	87	User	Epoetin alfa/beta	36	

DA, darbepoetin alfa; DD, dialysis dependent; ESA, erythropoiesis-stimulating agent; HD, hemodialysis; NDD, non-dialysis dependent; PD, peritoneal dialysis.

## Data Availability

Data sharing not applicable.

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
