# Peer review of "Hypoxia-Inducible Factor Prolyl Hydroxylase Inhibitors and Iron Metabolism"

_ijms, 2023, doi:10.3390/ijms24033037_

Round 1

Reviewer 1 Report

This is a well written review article that systematically summarized the molecular mechanism and clinical practice of HIF prolyl hydroxylase inhibitors in the treatment of renal anemia. 

Actually, findings from clinical practice summarized in the manuscript are trials were performed in Japan, thus, this information should be correspondingly mentioned in the abstract, e.g. “Clinical studies to date in Japan have also shown that HIF-PHIs not only promote hematopoiesis, but also improve iron metabolism ”. 

In Line 377,  the symbol “ ] “ at the end of the sentence, please confirm if any reference is missing.

The full name of CRP should be given when it was first mentioned in the manuscript. 

Author Response

This is a well written review article that systematically summarized the molecular mechanism and clinical practice of HIF prolyl hydroxylase inhibitors in the treatment of renal anemia. Actually, findings from clinical practice summarized in the manuscript are trials were performed in Japan, thus, this information should be correspondingly mentioned in the abstract, e.g. “Clinical studies to date in Japan have also shown that HIF-PHIs not only promote hematopoiesis, but also improve iron metabolism ”.

Thank you for your warm comments throughout. We have added “ in Japan” as follows.

Clinical studies to date in Japan have also shown that HIF-PHIs not only promote hematopoiesis, but also improve iron metabolism. (Line 24)

In Line 377,  the symbol “ ] “ at the end of the sentence, please confirm if any reference is missing.

Thank you for your suggestion.

We have removed the symbol “ ] “. (Line 387)

The full name of CRP should be given when it was first mentioned in the manuscript.

Thank you for your suggestion.

We have added the full name of CRP. (Line 327)

Reviewer 2 Report

Abstract

Differentiate between HIF-1 and HIF-2. HIF-2 seems to be the principal regulator of EPO transcription, but HIF-1 may have additional influences on erythroid maturation.

Better to say iron absorption is strictly controlled.

EPO production is not suppressed in renal anemia, rather there is a lack of stimulation of EPO synthesis.

Improve iron metabolism is imprecise terminology. Do you mean iron availability for hemoglobin production? If so, where does this increased availability occur? For example, is it increased absorption into the small intestine epithelial cells, export from intestinal epithelial cells to the portal plasma, uptake by erythroid progenitors, and/or uptake by erythroid mitochondria for insertion of iron into heme by ferrochelatase? Or does it relate to decreased storage of iron in macrophages and increased export by ferroportin from macrophages Into the plasma?

Main text

Problems in imprecise terminology and sometimes outdated references persist. I suggest eliminating “improve iron metabolism” and “impair iron metabolism” throughout the text and rather use more informative terminology.

Table 1. I could not figure out the clarifications for starting doses of Roxadustat and Molidustat. It would be good to give results for erythropoietin and transferrin measurements, both of which are HIF-regulated. Otherwise point out that TIBC reflects transferrin concentration.

Need to consider that both increased hypoxic response and iron deficiency lead to increase in TIBC/transferrin levels.

It may be good to start the Review with “3.5 Hematopoiesis and iron metabolism” and to incorporate accurate supportive information from earlier in the paper as needed in the text in an abbreviated form.

Author Response

Comments and Suggestions for Authors

Abstract

Differentiate between HIF-1 and HIF-2. HIF-2 seems to be the principal regulator of EPO transcription, but HIF-1 may have additional influences on erythroid maturation.

Thank you for your suggestion. We have added your suggestion as follows and moved one sentence forward in “ 4.2. HIF and hematopoiesis”

HIF2α seems to be the principal regulator of EPO transcription, but HIF1α and 3α may have additional influences on erythroid maturation. (Line 11-12)

HIF1α has been shown to induce the expression of GATA1, an important factor in the differentiation of erythroid cells, and to inhibit apoptosis of CFU-E and erythroblasts [100] (Line 246-247)

Better to say iron absorption is strictly controlled.

Thank you for your suggestion. We have added about iron absorption as follows.

so its absorption and retention are strictly controlled. (Line 14)

EPO production is not suppressed in renal anemia, rather there is a lack of stimulation of EPO synthesis.

Thank you for your suggestion. We have rewritten as you pointed out.

which there is a lack of stimulation of EPO synthesis due to decreased HIF expression. (Line 18)

Improve iron metabolism is imprecise terminology. Do you mean iron availability for hemoglobin production? If so, where does this increased availability occur? For example, is it increased absorption into the small intestine epithelial cells, export from intestinal epithelial cells to the portal plasma, uptake by erythroid progenitors, and/or uptake by erythroid mitochondria for insertion of iron into heme by ferrochelatase? Or does it relate to decreased storage of iron in macrophages and increased export by ferroportin from macrophages Into the plasma?

Thank you for telling us about Terminology. It is true that there are many ambiguous expressions and I thought that they do not accurately represent the physiology and pathophysiology. As you instructed, we have tried to be precise by using logical and scientific expressions. We have the impression that we were given a lecture at a university in the past, and we would like to appreciate you again.

As you pointed out, HIF-PHI may improve all of the above, since it exhibits HIF-1,2α effects. However, it would be too long for abstract to describe everything, and we have rewritten "iron metabolism" as follows to include the effects that you have pointed out.

Therefore, unlike erythropoiesis-stimulating agents, HIF-PHI may enhance iron absorption from the intestinal tract and iron supply from reticuloendothelial macrophages and hepatocytes into the plasma, thus facilitating the availability of iron for hematopoiesis. (Line 20-22)

Clinical studies to date in Japan have also shown that HIF-PHIs not only promote hematopoiesis, but also decrease hepcidin, the main regulator of iron metabolism, and increase total iron-binding capacity (TIBC), which indicates iron transport capacity. (Line 24-26)

Main text

Problems in imprecise terminology and sometimes outdated references persist. I suggest eliminating “improve iron metabolism” and “impair iron metabolism” throughout the text and rather use more informative terminology.

We have followed your instructions and revised the descriptions in the text to more accurately represent the condition, and we list the changes in each section. So, we have rewritten parts of “improve iron metabolism” and “impair iron metabolism” according to your suggestions as follows.

・HIF-PHIs also improve iron supply for hematopoiesis [10]. (Line 49)

・Many cases of ESA resistance are attributed to iron deficiency [16]. (Line 56)

・HIF-PHIs not only promote endogenous EPO production, but also improve iron supply for hematopoiesis by promoting iron absorption from the intestinal tract and iron recirculation, and thus have promise for improving ESA-resistant anemia caused by iron deficiency and reducing the use of IV iron therapy [10]. (Line 61-64)

・in fact clinical data have shown reduction in hepcidin and elevation in TIBC, which reflects Tf levels. Available iron deficiency is a common cause of ESA resistance [16], (Line 283-285)

・During inflammation, available iron deficiency due to increased hepcidin levels results in ESA resistance. HIF-PHIs may improve iron supply from macrophages and hepatocellular into the plasma even in ESA-resistant patients through actions such as lowering hepcidin levels. (Line 330-333)

・The results showed enhanced hematopoiesis, suppressed hepcidin, and increased TIBC, immediately after the switch, (Line 338-339)

We agree with your pointed out problem, that is “sometimes outdated references persist”.

So, the references on the basic factors of iron metabolism remains outdated, but we have replaced the old references of clinical studies and reviews with new ones as follows.

References

  1. Chen, T.K; Knicely, D.H; Grams, M.E. Chronic Kidney Disease Diagnosis and Management: A Review. JAMA. 2019, 322, 1294-1304. doi: 10.1001/jama.2019.14745.

  1. Hanna, R.M; Streja, ;, Kalantar-Zadeh, K. Burden of Anemia in Chronic Kidney Disease: Beyond Erythropoietin. Adv Ther. 2021, 38, 52-75. doi: 10.1007/s12325-020-01524-6.

  1. Nevo, A; Armaly, Z; Abd, E.l; Kadir, A; Douvdevani, A; Tovbin, D. Elevated Neutrophil Gelatinase Lipocalin Levels Are Associated With Increased Oxidative Stress in Hemodialysis Patients. J Clin Med Res. 2018, 10, 461-465. doi: 10.14740/jocmr3360w.

  1. Agarwal, R; Kusek, J.W; Pappas, M.K. A randomized trial of intravenous and oral iron in chronic kidney disease. Kidney Int 2015, 88, 905-914. doi: 10.1038/ki.2015.163.

  1. Piskin, E; Cianciosi, D; Gulec, S; Tomas, M; Capanoglu, E. Iron Absorption: Factors, Limitations, and Improvement Methods. ACS Omega. 2022, 7, 20441-20456. doi: 10.1021/acsomega.2c01833.

  1. Webster, A.C; Nagler, E.V; Morton, R.L; Masson, P. Chronic Kidney Disease. Lancet. 2017, 389, 1238-1252. doi: 10.1016/S0140-6736(16)32064-5.

Table 1. I could not figure out the clarifications for starting doses of Roxadustat and Molidustat. It would be good to give results for erythropoietin and transferrin measurements, both of which are HIF-regulated. Otherwise point out that TIBC reflects transferrin concentration.

Thank you for your suggestion.

We have rewritten starting dose and added the explanations due to find out the starting dose of Roxadustat and Molidustat and explanation of TIBC as follows. And units for iron-related factors in Table 1 have been removed as they are not given numerical values.

Roxadustat

*; independent of body weight (sorted by the web registration system), **; ESA dose before switching (DA < 20 μg/wk, CERA ≤ 100 µg/4 wk, rHuEPO < 4500 IU/w), ***; ESA dose before switching (DA ≥ 20 μg/wk, CERA > 100 µg/4 wk, rHuEPO ≥ 4500 IU/wk),

Molidustat

†; ESA dose before switching (DA ≤ 30 μg/4 wk, CERA 25 μg/4 wk, rHuEPO ≤ 3000/2 wks), ††; ESA dose before switching (DA > 30 μg/4 wk, CERA 25 μg/4 wk, rHuEPO > 3000/2 wks)

TIBC, total iron-binding capacity, which reflects transferrin levels

Unfortunately, no phase III trial measured erythropoietin over time. We have added transferrin in Tables 1 and 2 because it was measured in some HIF-PHI trials.

Need to consider that both increased hypoxic response and iron deficiency lead to increase in TIBC/transferrin levels.

Thank you for your suggestion. In our Roxadustat study, despite the low median serum ferritin level of 46.6 mg/dL, changes in TIBC were not related to changes in serum ferritin, and only changes in RBC showed a strong correlation. Therefore, we have added the following sentences.

TIBC is elevated not only by HIF1α but also by iron deficiency. However, in our study, despite the low median serum ferritin level of 46.6 mg/dL at baseline, the increase in TIBC was not associated with changes in serum ferritin. It positively correlated with only the change in red blood cell count [138], suggesting that the increase in TIBC may be due to the effect of HIF. (Line 341-345)

It may be good to start the Review with “3.5 Hematopoiesis and iron metabolism” and to incorporate accurate supportive information from earlier in the paper as needed in the text in an abbreviated form.

In accordance with your suggestion, we have placed “3.5 Hematopoiesis and iron metabolism” next to introduction and added or moved some sentences for clarity.(Line 68-96)

Also, Figure 4 was moved to 1.

Therefore, erythropoietin and iron, which is the main component of Hb, are essential for hematopoiesis. And erythropoietin production is mainly regulated by HIF2α [25]. (Line 78-80)

Recent evidence suggests that ERFE suppresses hepcidin production by directly acting on the BMP/SMAD system [47]. (From “Hematopoiesis and iron metabolism” to” 4. Iron metabolism, 4.1. General information”) (Line 146-147)
